# Intimate partner violence against women with disability and associated mental health concerns: a cross-sectional survey in Mumbai, India

Andrew Riley ,[1] Nayreen Daruwalla,[2] Suman Kanougiya ,[3] Apoorwa Gupta,[2] Mary Wickenden,[4] David Osrin [5]

[1]Institute of Population Health, University of Liverpool, Liverpool, UK
[2]Program on Prevention of Violence Against Women and Children, SNEHA, Mumbai, Maharashtra, India
[3]Tata Institute of Social Sciences (TISS), Mumbai, Maharashtra, India
[4]Institute of Development Studies, University of Sussex, Brighton, UK
[5]Institute for Global Health, University College London, London, UK

**Correspondence to**
Andrew Riley;
andrew.riley@liverpool.ac.uk

## ABSTRACT

**Objectives** The risk of intimate partner violence (IPV) against women with disability is believed to be high. We aimed to compare the prevalence of past-year IPV against women with and without functional difficulties in urban informal settlements, to review its social determinants and to explore its association with mental health.

**Design** Cross-sectional survey.

**Setting** Fifty clusters within four informal settlements.

**Participants** 5122 women aged 18–49 years.

**Primary and secondary outcome measures** We used the Washington Group Short Set of Questions to assess functional difficulties. IPV in the past year was described by binary composites of questions about physical, sexual and emotional violence. We screened for symptoms of depression using the Patient Health Questionnaire-9 and of anxiety using the Generalised Anxiety Disorder-7. Multivariable logistic regression models examined associations between functional difficulties, IPV and mental health.

**Results** 10% of participants who screened positive for functional disability had greater odds of experiencing physical or sexual IPV (adjusted OR (AOR) 1.68, 95% CI 1.23 to 2.29) and emotional IPV (1.52, 95% CI 1.16 to 2.00) than women who screened negative. Women who screened positive for functional disability had greater odds than women who screened negative of symptoms suggesting moderate or severe anxiety (AOR 2.50, 95% CI 1.78 to 3.49), depression (2.91, 95% CI 2.13 to 3.99) and suicidal thinking (AOR 1.94, 95% CI 1.50 to 2.50).

**Conclusions** The burden of IPV fell disproportionately on women with functional difficulties, who were also more likely to screen positive for common mental disorder. Public health initiatives need to respond at local and national levels to address the overlapping and mutually reinforcing determinants of violence, while existing policy needs to be better utilised to ensure protection for the most vulnerable.

## BACKGROUND

Approximately 15% of the world's population, around one billion people, live with a physical, intellectual, sensory or mental health impairment that affects their daily lives and long-term health and well-being.[1] Disability is

### Strengths and limitations of this study

► Data came from a large cross-sectional survey of women in informal settlements.
► Data were collected by an organisation providing support for survivors of intimate partner violence.
► Functional difficulties were self-reported and not the primary focus of the survey.
► The cross-sectional nature of the study limits the possibility of causal inference.

the product of attitudinal and environmental barriers that limit persons with functional difficulties from fully participating in society.[2] (In this paper, our approach to disability is in line with the social model which identifies disability as a social creation, 'a relationship between people with impairment and a disabling society'[3], distinct from medical and individual models of disability. We, therefore, use the term 'disabled women', which denotes this position, interchangeably with 'women with disabilities'). These barriers mean that, compared with non-disabled people, people with disability are less likely to be educated,[4] financially secure,[5] employed,[6] have access to health services,[1] have their health needs met[1 7] and have their social needs met.[8–10]

The double burden on women and girls with disability means that they are among the more marginalised and vulnerable in society, particularly when this burden intersects with multidimensional poverty.[1] Lacking physical and financial independence or access to socially inclusive services, disabled women who depend on partners and families for support are vulnerable to further social marginalisation and isolation and are at higher risk of physical violence and sexual abuse,[11 12] as well as other forms of abuse such as neglect, exploitation and coercive control.[13] Irrespective of disability, violence

against women occurs most commonly in the home, in the form of intimate partner violence (IPV) or non-partner domestic violence.[14 15] Women with disability are believed to be at greater risk of these and other forms of violence. However, a dearth of population-level research,[16] and the typically hidden nature of violence,[17] mean that little is known of how disability status may lead to additional risks of IPV. Such research is important for informing IPV prevention programmes,[18 19] and for meeting the objectives of the United Nations Sustainable Development Goals.

In India, the 2011 national census[20] and the 2018 National Sample Survey[21] both estimated disability prevalence at 2.2% (26.8 million people), although these are considered underestimates due to proxy reporting and insufficient consideration of disability.[22 23] Other national estimates range from 8% by the World Bank,[23] to 25% by the World Health Survey.[1 24] More local estimates reflect different approaches to classification and include 1.6% in Tamil Nadu State,[25] 1.9% in a Delhi slum community[26] and 12.2% in Telangana State.[27] Disability is most common among older, rural women and members of scheduled tribes and scheduled castes,[21 28] while prevalence increases with age, among lower-socioeconomic groups and those without employment/the unemployed.[25 27] Disabled women often report lower rates of marriage,[21 27] and higher rates of widowhood, separation and divorce.[21 25]

The ubiquity of violence against women in India has been well reported in both popular media and population-level research. The fourth National Family Health Survey (NFHS-4) reported that 30% of married women had experienced physical, sexual or emotional violence in their lifetime and 26% in the past year.[29] Similar rates were reported in a nationally representative systematic review[30] and in urban and slum settings.[31–35] Violence against women with disabilities in India is less documented: to date, only three major studies to date have reported on it at population level. A survey in Mumbai involving women aged 15–49 years with functional impairments (visual, locomotor, hearing) found that 20% of 123 ever-married disabled women had experienced past-year physical IPV, 23% emotional IPV (being insulted, humiliated or threatened) and 10% sexual IPV. Violence perpetrated by people other than intimate partners was experienced by 18% of ever-married and 23% of unmarried women, most commonly by mothers-in-law in the marital family and brothers in the natal.[12]

In Karnataka, of 70 female interviewees with disability, 73% reported 'significant' violence (vs 'rare' or 'occasional') and 23% sexual violence in the previous 12 months, while 59% reported generalised violence and 10% sexual violence in their childhood. Around half of this violence was perpetrated by family members (46%) or by people outside the family (54%).[36] In a study involving 729 women in Odisha state, 48% of those with learning difficulties and 23% with physical impairments reported experiencing domestic violence.[37] There are a number of limitations to these studies, notably their small sample sizes and, in the latter two, their imprecision in defining violence and the use of convenience sampling[36 37]; however, their findings are consistent with other reports from the wider South Asia region.[38–40]

For all women, violence increases the risk of serious injury and mortality,[41 42] HIV and sexually transmitted infections,[43 44] unintended pregnancy,[45] reproductive health morbidity[46 47] and a range of mental health concerns.[48–50] Among women with disabilities in India who have experienced violence, high levels of severe mental distress, depression, suicidal ideation and attempted suicide have been documented.[12]

Given its impact on physical and mental health, as well as being a matter of individual and social justice, violence against disabled women is largely absent from the literature. This absence limits the capacity of health planners and policymakers to act, while also contributing to the ongoing marginalisation of disabled communities, experiences and concerns.

## Objectives

This study sought to address the gap in understanding of the sociodemographic circumstances of women with disabilities in urban India and their experience of IPV. We aimed to identify associations between individual socioeconomic determinants and disability in women aged 18–49 years living in informal settlements in Mumbai and to test the hypothesis that women with functional difficulty/disability are more likely to have experienced recent IPV and have higher rates of common mental disorders.

## METHODS
### Setting

Maharashtra ranks fourth among Indian states in the human development index,[51] and women tend to have higher than the national averages on indicators of literacy, marriage age and workforce participation. Mumbai, Maharashtra's capital, has approximately 12.7 million residents, 41% of whom live in informal settlements.[52 53] These are characterised by overcrowding, lack of tenure and insufficient living space, housing durability and water and sanitation facilities.[54 55] Women in these disadvantaged communities typically depend on relatives for housing, making them vulnerable to homelessness, poverty and violence.[55]

### Design

We used data collected by (Society for Nutrition, Education and Health Action, a non-government organisation working in Mumbai's disadvantaged communities to improve the health of women and children and address the burden of violence. The dataset came from a survey conducted before the implementation of a community-based intervention to prevent violence against women and girls through individual volunteers and groups.[56]

The survey included 50 clusters of equal size, each of 500 dwellings, covering a population of 77 000 in four areas within major informal settlements.

## Participants and data collection

Data collection has been described in detail elsewhere.[57] Clusters were mapped and homes visited to identify potential participants. Criteria for inclusion were women aged 18–49 years resident in a study cluster, who consented to interview. One woman per household was sampled (around 100 participants per cluster). Data collectors visited participants to explain the survey, discuss the process, answer questions and provide a participant information sheet. Signed consent to interview was obtained and participants were assured of data confidentiality.

Sixteen women interviewers collected data. They were graduates who had not worked in the mental health field and were selected after application and interview through open recruitment. They received 3 months of training, which included 3 days of training facilitated by an expert on disability awareness and the Washington Group Short Set of Questions (WG-SS). For participants with visual impairment, the interviews were oral, as they were for all participants. For participants with hearing impairment, a communications specialist accompanied the interviewer. For participants with learning difficulties, we developed an alternative interview in a simple visual format that was used 14 times during the survey.

Participants were asked about demographic characteristics, socioeconomic indicators, their health, well-being, functional difficulty and disability status, common mental disorders, their experience of emotional, physical and sexual domestic violence, injuries, help-seeking, support and disclosure, and spousal drug and alcohol use. All participant information was confidential and no names or identifiable details appear in outputs or analyses. Sources of questions included the NFHS-4,[29] the Indian Family Violence and Control Scale,[58] the International Violence Against Women Survey[59] and the WHO multicountry study.[60] Questions were translated where necessary, back-translated, piloted and refined (see online supplemental file 1).

## Ethical considerations

The survey involved disclosing personal and sensitive information, which raises issues of interviewer behaviour, consent, data sharing, and privacy and confidentiality. All interviewers underwent appropriate training and WHO ethical and safety recommendations were followed.[61] Interview processes were checked through random visits and systematic monitoring by project officers and supervisors.

During the interview, participants were made aware of the potentially sensitive nature of the questions about violence and consent was obtained before they began. Efforts were made to ensure privacy by prearranging visits at a preferred time and location. Participants were able to attend a local community office, but most were interviewed at home. If household members were unwilling to leave, researchers encouraged them to listen to some of the routine questions to be reassured of the survey's nature. Most were amenable to leaving, but when they were not, the interview was terminated and completed over up to three repeat visits

Safeguarding was paramount during the survey. All participants were given information on local counselling and crisis support services for themselves or other women, whether they reported violence or not. Service providers were made aware of the study and the potential for increased service use. Researchers were able to arrange referrals if requested. For those who disclosed experience of violence, duty of care was of particular concern. Details of safeguarding measures and counselling services made available to survivors are described elsewhere.[57]

## Sample size

An estimate of prevalence from a cross-sectional sample of 5000 in a population of 125 000 would have a precision of around 1%. Within this, a comparison of two categories of determinant for 100 participants in each of 50 clusters would provide 80% power at 5% significance level to detect a difference of 6% in prevalence estimates of 10%–20%.

## Variables

Disability was assessed using the WG-SS, based on the International Classification of Functioning, Disability and Health model of disability.[62] The questions cover six functional domains (seeing, hearing, walking, memory, self-care and speaking), with four categories of response (no difficulty, some difficulty, a lot of difficulty and cannot do at all).[62] For this analysis, a participant was classified as having disability if they reported 'some difficulty' in at least one domain. This cut-off is one of several possible approaches and has been used in a similar setting.[38]

Definitions of violence follow the WHO Multi-Country Study[60] and are discussed in detail elsewhere.[57] We screened for anxiety, depression and suicidal thinking as common mental disorders. Anxiety was screened for using the Generalised Anxiety Disorder 7-item scale.[63] Depression was screened for using the Patient Health Questionnaire-9.[64] For both instruments, a cut-off score of ≥10 has been used as suggestive of clinically relevant conditions requiring further assessment.[65] In this study, variables were coded 'yes' for depression and anxiety if responses were recorded as moderate or severe. Suicidal thinking was coded 'yes' for lifetime positive responses.

Covariates corresponded to determinants of risk for IPV identified from the wider body of Indian IPV research undertaken in comparable settings with women whose disability status was not identified. Covariates were classified by level of occurrence, following Heise's ecological framework for violence against women.[66 67] This approach conceptualises risk of violence as a function of mutually interacting personal, situational and sociocultural factors

across individual, relationship, household, community, and society levels.

We hypothesised that a woman's odds of experiencing IPV would increase with age,[68–70] number of children[41 68] and employment outside the home,[71–73] and would decrease with years of education.[33 41 74] Partner use of alcohol or drugs was hypothesised to increase the odds,[31 69 75–77] while no clear relationship between IPV and partner level of education was predicted.[34 68 78 79] Lower socioeconomic status was considered as potentially increasing the odds of IPV,[12 34 35 41 76 80] as were faith[69 80] and caste,[69 81] due to the association of some groups with lower socioeconomic status.

Associations between common mental disorders and sociodemographic characteristics were hypothesised to conform to a similar pattern due to their close association with IPV. Hypothetical determinants were modelled as independent variables in both univariable and multivariable analyses.[82] At the level of individual women, age group, marital status, parity, education and employment were entered as categorical variables. Categories of age and schooling were based on those used in other studies.[38 41 43 46] Individual employment categories were recoded as either home-based or outside the home as IPV has been associated with women working outside the home in general rather than individual employment categories.[71–73] At 'partner' level, husband's schooling and employment were entered as categorical indicators and alcohol or drug use as binary. At 'household' level, faith, caste and socioeconomic quintile were entered as categorical indicators. Socioeconomic quintile was entered as a categorical rather than continuous variable to retain categorical consistency and because it was hypothesised that a reduction in odds would be observed in the wealthier quintiles. Quintiles were based on analysis of 22 individual household assets, with scores derived from standardised weights from the first component of a principal components analysis.[83 84]

### Statistical analysis

The survey had a single-stage design with clustering and stratification. Data analyses were unweighted because the clusters were of similar size and each was sampled for about 100 questionnaires. Sociodemographic factors were summarised and cross-tabulated using frequencies and percentages, and univariable logistic regression models used to test associations between each exposure variable and the outcome. A total of 204 values were missing in the socioeconomic quintile variable and 317 in variables relating to husbands, which were accounted for in the regression analysis through listwise deletion. Variables with associations at p<0.1 were included in multivariable analysis. Multivariable logistic regression was used to examine associations between outcome variables (disability, IPV, depression, anxiety) and primary exposure variables, adjusting for confounders (individual demographic and socioeconomic characteristics). Post hoc Wald tests were performed to check models. All

analyses used survey commands in Stata V.15.1. The data are available online at Open Science Framework,[85] available from https://osf.io/zhtpw/.

### Patient and public involvement

Our research responds to the urgency of preventing violence and improving services for survivors, primarily through community-based programming in informal settlements. It is driven by survivors' needs, one of which is inclusion. We followed a protocol co-developed with community members to achieve maximal inclusion of women with functional difficulties. We discussed the survey questions with participants in a pilot phase and adapted them for subsequent use. Our findings will be shared through community women's groups, three of which meet monthly in each of the 50 informal settlement clusters.

## RESULTS

Interviewers approached 5277 households between 5 December 2017 and 28 March 2019. In 967 (5%) households, there was no eligible female resident; in 592 households (11%) the interviewers could not achieve privacy across repeat visits; and in 155 (3%) households a potential participant declined interview. In total, 5122 interviews were conducted.

### Disability prevalence

Table 1 summarises the type and frequency of functional difficulties reported by women. Ten per cent had 'some difficulty' in at least one domain, of which the most common were visual (4%) and locomotor (5%). Across all domains, the number of participants reporting a lot of difficulty or no ability at all was low, each less than 1%.

### Women's sociodemographic profiles and disability

Most women were in their 20s or 30s (80%), currently married (92%) and with one or two children (51%). One-fifth of women (19%) reported having no schooling, while 44% had had 6–10 years. Having no remunerated work was common (76%), while those who had an income tended to work from home (15%). In unadjusted analyses, greater odds of disability were observed in women over 40 (OR 2.7; 95% CI 1.3 to 5.5), women who were separated or divorced (3.3; 95% CI 1.9 to 5.6), or widowed (3.6; 95% CI 2.2 to 5.9) and women who had three or more children (1.9; 95% CI 1.4 to 2.4). Attending school for at least 6 years was associated with lesser odds of disability than no schooling at all (0.8; 95% CI 0.6 to 0.9). Women who earned in the home had greater odds of disability than women who did not (1.5; 95% CI 1.1 to 2.0), as did women from Muslim rather than Hindu families (1.4; 95% CI 1.1 to 1.8).

### Associations between disability and past-year IPV

Table 2 presents associations between women's functional difficulty and past-year physical, sexual and emotional IPV. Of 5122 women, 628 reported suffering physical or

**Table 1** Frequencies of domains of functional difficulty among 5122 women aged 18–49 years living in informal settlements in Mumbai

|  | None | Some difficulty | A lot of difficulty | Cannot do at all |
|---|---|---|---|---|
| **Functional difficulty** | **n (%)** | **n (%)** | **n (%)** | **n (%)** |
| Seeing | 4896 (96) | 212 (4) | 12 (<1) | 2 (<1) |
| Hearing | 5078 (99) | 40 (<1) | 3 (<1) | 1 (<1) |
| Walking | 4860 (95) | 234 (5) | 25 (<1) | 3 (<1) |
| Memory | 4988 (97) | 123 (2) | 11 (<1) | 0 (0) |
| Self-care | 5108 (>99) | 10 (<1) | 4 (<1) | 0 (0) |
| Speaking | 5109 (>99) | 4 (<1) | 9 (<1) | 0 (0) |
| **Washington group disability prevalence assessment** | | | | **n (%)** |
| At least *one domain some difficulty* | | | | 505 (10) |
| At least one domain a lot of difficulty | | | | 63 (1) |
| At least one domain cannot at all | | | | 6 (<1) |
| At least one domain a lot of difficulty or cannot at all or at least some difficulty in two domains | | | | 178 (3) |

sexual IPV in the last year (12%) and 607 had suffered emotional IPV (12%). These proportions were greater for women with disability: 84 of 505 who reported at least some functional difficulty had suffered physical or sexual IPV (17%) and 83 emotional IPV (16%). We had information on socioeconomic asset scores for 4918 women who had ever been married, and on the husbands of 4805 women whose husbands were alive. After adjustment for age, marital status, parity, woman's and husband's schooling, woman's and husband's employment, religion and husband's use of alcohol or drugs, these differences were manifest in adjusted ORs (AOR) of 1.68 (95% CI 1.23 to 2.29) for physical or sexual and 1.52 (95% CI 1.16 to 2.00) for emotional IPV.

### Associations between disability and symptoms of anxiety and depression

Table 3 compares the profiles of women screened as having symptoms of at least moderate anxiety or depression with those who did not. 10% of women fell into this category and disabled women had nearly three times greater odds of doing so than non-disabled women (AOR 2.88, 95% CI 2.17 to 3.82). Women who reported physical or sexual IPV (AOR 2.50, 95% CI 1.77 to 3.53) or emotional IPV (AOR 2.34, 95% CI 1.59 to 3.43) also had greater odds of anxiety or depression. Associations with anxiety or depression were also seen for separated, divorced or widowed women, who were engaged in remunerated work, and whose husbands used alcohol or drugs. Associations were not seen in unadjusted models for parity or caste.

Table 4 examines associations in more detail, presenting ORs for positive screens for moderate or severe anxiety, depression and suicidal thinking in the presence of disability and physical, sexual or emotional IPV. Overall, 6% of women screened positive for anxiety symptoms and 9% for symptoms of depression, while 13% had experienced suicidal thinking. In unadjusted

models, disability increased the odds of each of these by two to four times. Adjusting for potential confounders reduced the ORs, but they remained substantial. IPV may also be a confounding factor, as it is associated with both mental health concerns and disability. Adjusting for IPV and sociodemographic characteristics further reduced the ORs, but women with disability continued to have elevated odds of having a mental health concern. In this second adjusted model, women with functional difficulty had more than twice the odds of reporting anxiety symptoms as non-disabled women (AOR 2 2.50, 95% CI 1.78 to 3.49) and nearly three times the odds of depression (AOR 2 2.91, 95% CI 2.13 to 3.99). They had almost twice the odds of reporting suicidal thinking (AOR 2 1.94, 95% CI 1.50 to 2.50). Independent of functional status, physical, sexual and emotional IPV were all associated with greater odds of having a mental health concern.

### DISCUSSION

In a cross-sectional survey of women living in informal settlements in Mumbai, disability was more common in women over 30 years of age and with children, who were separated, divorced or widowed, and from poorer households. Women with disability had over 50% greater odds of reporting physical, sexual or emotional IPV than non-disabled women, and disabled women and those who had experienced IPV had greater odds of anxiety, depression and suicidal thinking. Overall, the increased odds of disabled women experiencing IPV are consistent with studies from high-income[13 86] and low-income settings.[38 87] These findings contribute to our understanding of a serious but insufficiently investigated public health issue.[16]

The reported prevalence of IPV among all women (12% for physical or sexual and emotional IPV) was lower than at national level[29 30] and in slums.[33 34] It was

**Table 2** Associations between experience of intimate partner violence (physical or sexual IPV, emotional IPV) in the preceding year, disability and sociodemographic characteristics, for 5122 women aged 18–49 years living in informal settlements in Mumbai

| Characteristic (n) | Physical or sexual IPV in last year | | | | | | | Emotional IPV in last year | | | | | | |
|---|---|---|---|---|---|---|---|---|---|---|---|---|---|---|
| | No n (%) | Yes n (%) | OR | (95% CI) | P value | Adjusted OR | (95% CI) | No n (%) | Yes, n (%) | OR | (95% CI) | P value | Adjusted OR | (95% CI) |
| Disability | | | | | | | | | | | | | | |
| None | 4073 (88) | 544 (12) | 1 | | | 1 | | 4093 (89) | 524 (11) | 1 | | | 1 | |
| At least some difficulty | 421 (83) | 84 (17) | 1.49 | 1.24 to 1.99 | 0.007 | 1.68 | 1.23 to 2.29 | 422 (84) | 83 (16) | 1.54 | 1.20 to 1.97 | 0.001 | 1.52 | 1.16 to 2.00 |
| All | 4494 (88) | 628 (12) | | | | | | 4515 (88) | 607 (12) | | | | | |
| Woman's age (years) | | | | | | | | | | | | | | |
| <20 | 104 (90) | 11 (10) | 1 | | | 1 | | 108 (94) | 7 (6) | 1 | | | 1 | – |
| 20–29 | 1747 (86) | 291 (14) | 1.57 | 0.54 to 4.64 | | 0.82 | 0.26 to 2.56 | 1782 (87) | 256 (13) | 2.22 | 0.80 to 6.16 | | – | – |
| 30–39 | 1808 (88) | 248 (12) | 1.3 | 0.43 to 3.88 | | 0.57 | 0.18 to 1.86 | 1812 (88) | 243 (12) | 2.07 | 0.74 to 5.81 | | – | – |
| >40 | 835 (91) | 78 (9) | 0.88 | 0.28 to 2.75 | <0.001 | 0.39 | 0.11 to 1.36 | 812 (89) | 101 (11) | 1.92 | 0.69 to 5.34 | 0.329 | – | – |
| All | 4494 (88) | 628 (12) | | | | | | 4515 (88) | 607 (12) | | | | | |
| Marital status | | | | | | | | | | | | | | |
| Married | 4112 (87) | 593 (13) | 1 | | | 1 | | 4131 (88) | 574 (12) | 1 | | | 1 | |
| Unmarried | 203 (99) | 2 (1) | 0.07 | 0.02 to 0.27 | | 0.07 | 0.02 to 0.23 | 204 (99) | 1 (<1) | 0.04 | 0.01 to 0.26 | | 0.03 | 0.01 to 0.24 |
| Separated or divorced | 78 (78) | 22 (22) | 1.96 | 1.20 to 3.18 | | 1.44 | 0.88 to 2.33 | 71 (71) | 29 (29) | 2.94 | 1.90 to 4.54 | | 2.14 | 1.37 to 3.35 |
| Widowed | 101 (90) | 11 (10) | 0.76 | 0.44 to 1.29 | 0.001 | 0.79 | 0.45 to 1.41 | 109 (97) | 3 (3) | 0.2 | 0.07 to 0.59 | <0.001 | 0.14 | 0.04 to 0.44 |
| All | 4494 (88) | 628 (12) | | | | | | 4515 (88) | 607 (12) | | | | | |
| Parity | | | | | | | | | | | | | | |
| No children | 692 (92) | 63 (8) | 1 | | | 1 | | 692 (92) | 63 (8) | 1 | | | 1 | |
| 1 or 2 | 2263 (86) | 358 (14) | 1.74 | 1.29 to 2.34 | | 1.41 | 1.04 to 1.91 | 2279 (87) | 342 (13) | 1.65 | 1.28 to 2.13 | | 1.16 | 0.89 to 1.51 |
| 3 or more | 1539 (88) | 207 (12) | 1.48 | 1.09 to 2.00 | 0.002 | 1.29 | 0.92 to 1.80 | 1544 (88) | 202 (12) | 1.44 | 1.09 to 1.90 | 0.002 | 0.96 | 0.70 to 1.31 |
| All | 4494 (88) | 628 (12) | | | | | | 4515 (88) | 607 (12) | | | | | |
| Woman's schooling | | | | | | | | | | | | | | |
| None | 848 (89) | 100 (11) | 1 | | | 1 | | 854 (90) | 94 (10) | 1 | | | 1 | |
| 1–5 years | 734 (85) | 125 (15) | 1.44 | 1.03 to 2.02 | | 1.39 | 0.99 to 1.96 | 737 (86) | 122 (14) | 1.5 | 1.13 to 2.00 | | 1.45 | 1.09 to 1.94 |
| 6–10 years | 1976 (87) | 304 (13) | 1.3 | 1.00 to 1.70 | | 1.24 | 0.93 to 1.66 | 1983 (87) | 297 (13) | 1.36 | 1.07 to 1.73 | | 1.29 | 1.00 to 1.67 |
| 11 years or more | 936 (90) | 99 (10) | 0.9 | 0.66 to 1.21 | 0.027 | 0.93 | 0.67 to 1.29 | 941 (91) | 94 (9) | 0.91 | 0.72 to 1.15 | 0.002 | 0.95 | 0.72 to 1.24 |
| All | 4494 (88) | 628 (12) | | | | | | 4515 (88) | 607 (12) | | | | | |
| Woman's employment | | | | | | | | | | | | | | |

Continued

**Table 2** Continued

| Characteristic (n) | Physical or sexual IPV in last year | | | | | | | Emotional IPV in last year | | | | | | |
|---|---|---|---|---|---|---|---|---|---|---|---|---|---|---|
| | No n (%) | Yes n (%) | OR | (95% CI) | P value | Adjusted OR | (95% CI) | No n (%) | Yes, n (%) | OR | (95% CI) | P value | Adjusted OR | (95% CI) |
| None | 3437 (89) | 433 (11) | 1 | | | 1 | | 3446 (89) | 424 (11) | 1 | | | 1 | |
| Home-based | 665 (86) | 108 (14) | 1.29 | 1.07 to 1.56 | | 1.22 | 1.01 to 1.49 | 677 (88) | 96 (12) | 1.15 | 0.92 to 1.44 | | 1.08 | 0.85 to 1.37 |
| Outside the home | 392 (82) | 87 (18) | 1.76 | 1.33 to 2.32 | <0.001 | 2.11 | 1.60 to 2.80 | 392 (82) | 87 (18) | 1.8 | 1.31 to 2.49 | 0.003 | 2 | 1.43 to 2.80 |
| All | 4494 (88) | 628 (12) | | | | | | 4515 (88) | 607 (12) | | | | | |
| Religion | | | | | | | | | | | | | | |
| Hindu | 2685 (89) | 317 (11) | 1 | | | 1 | | 2677 (89) | 325 (11) | 1 | | | 1 | |
| Muslim | 1629 (86) | 273 (14) | 1.42 | 1.15 to 1.76 | | 1.38 | 1.11 to 1.72 | 1655 (87) | 247 (13) | 1.23 | 1.00 to 1.52 | | 1.23 | 1.00 to 1.54 |
| Buddhist | 163 (83) | 34 (17) | 1.77 | 1.25 to 2.50 | | 1.68 | 1.16 to 2.43 | 167 (85) | 30 (15) | 1.48 | 1.11 to 1.97 | | 1.33 | 1.00 to 1.79 |
| Other | 17 (81) | 4 (19) | 1.99 | 0.64 to 6.23 | 0.003 | 1.9 | 0.63 to 5.74 | 16 (76) | 5 (24) | 2.57 | 0.90 to 7.36 | 0.013 | 2.94 | 0.93 to 9.27 |
| All | 4494 (88) | 628 (12) | | | | | | 4515 (88) | 607 (12) | | | | | |
| Caste | | | | | | | | | | | | | | |
| General | 2637 (88) | 350 (12) | 1 | | | – | – | 2651 (89) | 336 (11) | 1 | | | – | – |
| Other Backward Caste | 1064 (87) | 157 (13) | 1.11 | 0.92 to 1.35 | | – | – | 1063 (87) | 158 (13) | 1.17 | 0.95 to 1.45 | | – | – |
| Scheduled Caste/Tribe | 621 (86) | 103 (14) | 1.25 | 1.01 to 1.54 | | – | – | 629 (87) | 95 (13) | 1.19 | 0.93 to 1.53 | | – | – |
| None stated | 172 (91) | 18 (9) | 0.79 | 0.45 to 1.37 | 0.202 | – | – | 172 (91) | 18 (9) | 0.83 | 0.47 to 1.45 | 0.375 | – | – |
| All | 4494 (88) | 628 (12) | | | | | | 4515 (88) | 607 (12) | | | | | |
| Socioeconomic quintile (n=4918 ever-married women) | | | | | | | | | | | | | | |
| 1 poorest | 829 (84) | 161 (16) | 1 | | | 1 | | 849 (86) | 141 (14) | 1 | | | – | – |
| 2 | 861 (88) | 117 (12) | 0.7 | 0.55 to 0.88 | | 0.7 | 0.54 to 0.90 | 865 (88) | 113 (12) | 0.79 | 0.58 to 1.07 | | – | – |
| 3 | 865 (88) | 118 (12) | 0.7 | 0.52 to 0.95 | | 0.72 | 0.53 to 0.99 | 871 (89) | 112 (11) | 0.77 | 0.53 to 1.13 | | – | – |
| 4 | 876 (89) | 108 (11) | 0.63 | 0.49 to 0.83 | | 0.66 | 0.50 to 0.88 | 876 (89) | 108 (11) | 0.74 | 0.53 to 1.04 | | – | – |
| 5 least poor | 884 (90) | 99 (10) | 0.58 | 0.41 to 0.81 | 0.007 | 0.63 | 0.45 to 0.89 | 875 (89) | 108 (11) | 0.74 | 0.53 to 1.04 | 0.325 | – | – |
| All | 4315 (88) | 603 (12) | | | | | | 4336 (88) | 582 (12) | | | | | |
| Husband's schooling (n=4805 women with living husbands) | | | | | | | | | | | | | | |
| None | 391 (81) | 90 (19) | 1 | | | 1 | | 396 (82) | 85 (18) | 1 | | | 1 | |
| 1–5 years | 507 (86) | 83 (14) | 0.71 | 0.51 to 1.00 | | 0.73 | 0.51 to 1.04 | 509 (86) | 81 (14) | 0.74 | 0.50 to 1.10 | | 0.76 | 0.51 to 1.11 |
| 6–10 years | 2206 (87) | 318 (13) | 0.63 | 0.48 to 0.82 | | 0.66 | 0.50 to 0.87 | 2214 (88) | 310 (12) | 0.65 | 0.48 to 0.89 | | 0.68 | 0.50 to 0.92 |
| 11 years or more | 1069 (89) | 123 (11) | 0.54 | 0.39 to 0.75 | | 0.63 | 0.41 to 0.97 | 1066 (89) | 126 (11) | 0.59 | 0.43 to 0.83 | | 0.68 | 0.49 to 0.95 |
| Unknown | 17 (91) | 1 (9) | 0.44 | 0.29 to 0.66 | <0.001 | 0.55 | 0.37 to 0.84 | 17 (91) | 1 (9) | 0.48 | 0.33 to 0.70 | 0.005 | 0.6 | 0.40 to 0.92 |

Continued

**Table 2** Continued

| Characteristic (n) | Physical or sexual IPV in last year | | | | | | | Emotional IPV in last year | | | | | | |
| | No n (%) | Yes n (%) | OR | (95% CI) | P value | Adjusted OR | (95% CI) | No n (%) | Yes, n (%) | OR | (95% CI) | P value | Adjusted OR | (95% CI) |
|---|---|---|---|---|---|---|---|---|---|---|---|---|---|---|
| All | 4190 (87) | 615 (13) | | | | | | 4202 (87) | 603 (13) | | | | | |
| Husband employed in previous 12 months (n=4805 women with living husbands) | | | | | | | | | | | | | | |
| No | 73 (74) | 26 (26) | 1 | | | 1 | | 76 (77) | 23 (23) | 1 | | | 1 | |
| Yes | 4111 (88) | 586 (12) | 0.4 | 0.25 to 0.64 | | 0.5 | 0.29 to 0.84 | 4120 (88) | 577 (12) | 0.46 | 0.28 to 0.76 | | 0.75 | 0.45 to 1.27 |
| Unknown | 6 (67) | 3 (33) | 1.4 | 0.35 to 5.60 | 0.001 | 1.2 | 0.21 to 6.91 | 6 (67) | 3 (33) | 1.65 | 0.32 to 8.42 | 0.001 | 1.34 | 0.34 to 5.27 |
| All | 4190 (87) | 615 (13) | | | | | | 4202 (87) | 603 (13) | | | | | |
| Husband uses alcohol/drugs (n=4805 women with living husbands) | | | | | | | | | | | | | | |
| No | 3198 (89) | 377 (11) | 1 | | | 1 | | 3216 (90) | 359 (10) | 1 | | | 1 | |
| Yes | 992 (81) | 238 (19) | 2.04 | 1.59 to 2.61 | <0.001 | 1.81 | 1.36 to 2.40 | 986 (80) | 244 (20) | 2.22 | 1.79 to 2.75 | <0.001 | 2 | 1.59 to 2.53 |
| All | 4190 (87) | 615 (13) | | | | | | 4202 (87) | 603 (13) | | | | | |

Multivariable models (adjusted OR) include all variables for which adjusted ORs are presented. Of 5122 women, we had data on socioeconomic status for the 4918 who had ever been married and data on their husbands for 4805 who had not been widowed.
ORs and 95% CIs from univariable and multivariable logistic regression models.
CI, confidence interval; IPV, intimate partner violence; OR, odds ratio.

**Table 3** Associations of positive screens for moderate or severe anxiety (GAD-7) or depression (PHQ-9) with disability and intimate partner violence (IPV) in the preceding year, for 5122 women aged 18–49 years living in informal settlements in Mumbai

| Characteristic (n) | No, minimal or mild anxiety or depression n (%) | Moderate or severe anxiety or depression n (%) | OR | (95% CI) | P value | Adjusted OR | (95% CI) |
|---|---|---|---|---|---|---|---|
| **Disability** | | | | | | | |
| None | 4217 (91) | 400 (9) | 1 | | | 1 | |
| At least some impairment | 372 (74) | 133 (26) | 3.77 | 2.99 to 4.75 | <0.001 | 2.88 | 2.17 to 3.82 |
| All | 4589 (90) | 533 (10) | | | | | |
| **Physical or sexual IPV** | | | | | | | |
| No | 4132 (92) | 362 (8) | 1 | | | 1 | |
| Yes | 457 (73) | 171 (27) | 4.27 | 3.35 to 5.45 | <0.001 | 2.50 | 1.77 to 3.53 |
| All | 4589 (90) | 533 (10) | | | | | |
| **Emotional IPV** | | | | | | | |
| No | 4150 (92) | 365 (8) | 1 | | | 1 | |
| Yes | 439 (72) | 168 (28) | 4.35 | 3.37 to 5.62 | <0.001 | 2.34 | 1.59 to 3.43 |
| All | 4589 (90) | 533 (10) | | | | | |
| **Woman's age (years)** | | | | | | | |
| <20 | 102 (89) | 13 (11) | 1 | | | 1 | |
| 20–29 | 1854 (91) | 184 (9) | 0.78 | 0.44 to 1.38 | | 0.69 | 0.38 to 1.24 |
| 30–39 | 1857 (90) | 199 (10) | 0.84 | 0.47 to 1.51 | | 0.70 | 0.38 to 1.28 |
| >40 | 776 (85) | 137 (15) | 1.39 | 0.75 to 2.56 | <0.001 | 0.95 | 0.50 to 1.82 |
| All | 4589 (90) | 533 (10) | | | | | |
| **Marital status** | | | | | | | |
| Married | 4267 (91) | 438 (9) | 1 | | | 1 | |
| Unmarried | 189 (92) | 16 (8) | 0.82 | 0.46 to 1.48 | | 1.21 | 0.64 to 2.25 |
| Separated or divorced | 57 (57) | 43 (43) | 7.35 | 4.72 to 11.45 | | 4.73 | 2.68 to 9.34 |
| Widowed | 76 (68) | 36 (32) | 4.61 | 2.91 to 7.33 | <0.001 | 3.25 | 1.72 to 6.12 |
| All | 4589 (90) | 533 (10) | | | | | |
| **Woman's schooling** | | | | | | | |
| None | 819 (86) | 129 (14) | 1 | | | 1 | |
| 1–5 years | 758 (88) | 101 (12) | 0.85 | 0.61 to 1.17 | | 0.72 | 0.50 to 1.06 |
| 6–10 years | 2046 (90) | 234 (10) | 0.73 | 0.56 to 0.94 | | 0.70 | 0.50 to 0.96 |
| 11 years or more | 966 (93) | 69 (7) | 0.45 | 0.32 to 0.65 | <0.001 | 0.59 | 0.39 to 0.90 |
| All | 4589 (90) | 533 (10) | | | | | |
| **Woman's employment** | | | | | | | |
| None | 3535 (91) | 335 (9) | 1 | | | 1 | |
| Home based | 662 (86) | 111 (14) | 1.77 | 1.39 to 2.25 | | 1.48 | 1.12 to 1.96 |
| Outside the home | 392 (82) | 87 (18) | 2.34 | 1.73 to 3.18 | <0.001 | 1.70 | 1.19 to 2.44 |
| All | 4589 (90) | 533 (10) | | | | | |
| **Religion** | | | | | | | |
| Hindu | 2737 (91) | 265 (9) | 1 | | | 1 | |
| Muslim | 1669 (88) | 233 (12) | 1.44 | 1.15 to 1.81 | | 1.28 | 0.98 to 1.67 |
| Buddhist | 165 (84) | 32 (16) | 2.00 | 1.27 to 3.17 | | 1.63 | 0.98 to 2.70 |
| Other | 18 (86) | 3 (14) | 1.72 | 0.45 to 6.61 | 0.003 | 1.13 | 0.40 to 3.22 |
| All | 4589 (90) | 533 (10) | | | | | |
| **Socioeconomic quintile (n=4918 ever-married women)** | | | | | | | |

**Table 3** Continued

| Characteristic (n) | No, minimal or mild anxiety or depression n (%) | Moderate or severe anxiety or depression n (%) | OR | (95% CI) | P value | Adjusted OR | (95% CI) |
|---|---|---|---|---|---|---|---|
| 1 poorest | 857 (87) | 133 (13) | 1 | | | 1 | |
| 2 | 877 (90) | 101 (10) | 0.74 | 0.56 to 0.98 | | 0.73 | 0.53 to 1.02 |
| 3 | 879 (89) | 104 (11) | 0.76 | 0.56 to 1.04 | | 0.77 | 0.54 to 1.11 |
| 4 | 891 (91) | 93 (9) | 0.67 | 0.50 to 0.90 | | 0.68 | 0.48 to 0.96 |
| 5 least poor | 903 (92) | 80 (8) | 0.57 | 0.41 to 0.80 | 0.014 | 0.64 | 0.43 to 0.96 |
| All | 4407 (90) | 511 (10) | | | | | |
| Husband's schooling (n=4805 women with living husbands) | | | | | | | |
| None | 410 (85) | 71 (15) | 1 | | | 1 | |
| 1–5 years | 517 (88) | 73 (12) | 0.82 | 0.59 to 1.13 | | 1.04 | 0.76 to 1.44 |
| 6–10 years | 2282 (90) | 242 (10) | 0.61 | 0.44 to 0.84 | | 1.01 | 0.73 to 1.38 |
| 11 years or more | 1103 (91) | 89 (9) | 0.58 | 0.38 to 0.91 | | 1.19 | 0.72 to 1.95 |
| Unknown | 12 (94) | 6 (6) | 0.37 | 0.24 to 0.58 | <0.001 | 0.84 | 0.47 to 1.50 |
| All | 4324 (90) | 481 (10) | | | | | |
| Husband employed in previous 12 months (n=4805 women with living husbands) | | | | | | | |
| No | 72 (73) | 27 (27) | 1 | | | 1 | |
| Yes | 4247 (90) | 450 (10) | 0.28 | 0.19 to 0.42 | | 0.81 | 0.45 to 1.48 |
| Unknown | 5 (56) | 4 (44) | 2.13 | 0.49 to 9.23 | <0.001 | 1.70 | 0.29 to 9.94 |
| All | 4324 (90) | 481 (10) | | | | | |
| Husband uses alcohol or drugs (n=4805 women with living husbands) | | | | | | | |
| No | 3317 (93) | 258 (7) | 1 | | | 1 | |
| Yes | 1007 (82) | 223 (18) | 2.85 | 2.19 to 3.70 | <0.001 | 1.98 | 1.48 to 2.64 |
| All | 4324 (90) | 481 (10) | | | | | |

Multivariable models (adjusted OR) include all variables for which adjusted ORs are presented. Of 5122 women, we had data on socioeconomic status for the 4918 who had ever been married, and data on their husbands for 4805 who had not been widowed.
ORs and 95% CIs from univariable and multivariable logistic regression models.
CI, confidence interval; OR, odds ratio.

also lower among disabled women (17% for physical or sexual IPV, 16% for emotional IPV) compared with other studies.[12 36 37] The cause of this disparity is unclear. Strong efforts were made to help participants feel able to respond openly to questions, and the study used definitions of violence and survey instruments comparable with other studies: the lower rates may therefore be a fair reflection of levels of violence in this community. Indeed, the rate accords closely with a study of physical (12%), sexual (2%) and emotional IPV (8%) experienced 6 weeks post partum also undertaken in Mumbai informal settlements.[76] It is possible, nevertheless, that lower rates of violence may reflect under-reporting. Multiple factors play a role in the reporting of violence, such as a community's redressal structures, the extent of local welfare activities and social norms. It may be that some communities are particularly close and not forthcoming enough to report violence. The context of each community is different and efforts must be made to understand local norms and behaviours in order to contextualise research outcomes.

The mental health burden in India is high and greater treatment provision is needed.[88 89] One-in-seven Indian people are believed to be affected by some mental disorder and the burden of depressive and anxiety disorders is believed to be greatest among women.[90] For women with disabilities, the burden may be higher.[91] Two studies from South India found that disabled women were more likely to experience comorbid depression and diabetes than non-disabled women.[92 93] One of these found that disabled women were nearly ten times more likely to have depression than non-disabled women (AOR 9.5, 95% CI 2.2 to 40.8).[92] Our study supports these findings: disabled women had more mental health concerns even when controlling for socioeconomic conditions and experience of violence. Whether poor mental health and disability precede violence or vice-versa cannot be determined here. Comorbid mental health disorders may also result from pre-existing health conditions and be exacerbated by exposure to violence. These issues should be the subject of future research. Moreover, as with the elevated vulnerability to violence, this burden of mental disorder

**Table 4** Unadjusted and adjusted logistic regression models for associations between disability status, symptoms of anxiety and depression, and suicidal thinking, for 5122 women aged 18–49 years living in informal settlements in Mumbai

| | Moderate or severe anxiety n (%) | | OR | (95% CI) | Adjusted OR 1 | (95% CI) | Adjusted OR 2 | (95% CI) |
|---|---|---|---|---|---|---|---|---|
| | No | Yes | | | | | | |
| Disability status | | | | | | | | |
| None | 4389 (95) | 228 (5) | 1 | | 1 | | 1 | |
| At least some difficulty | 425 (84) | 80 (16) | 3.62 | 2.77 to 4.73 | 2.75 | 2.01 to 3.76 | 2.50 | 1.78 to 3.49 |
| All | 4814 (94) | 308 (6) | | | | | | |
| Physical or sexual IPV | | | | | | | | |
| No | 4293 (95) | 201 (4) | 1 | | 1 | | | |
| Yes | 521 (83) | 107 (17) | 4.39 | 3.41 to 5.65 | 4.05 | 2.98 to 5.50 | – | – |
| All | 4814 (94) | 308 (6) | | | | | | |
| Emotional IPV | | | | | | | | |
| No | 4315 (96) | 200 (4) | 1 | | 1 | | | |
| Yes | 499 (82) | 108 (18) | 4.67 | 3.59 to 6.07 | 4.11 | 3.08 to 5.49 | – | – |
| All | 4814 (94) | 308 (6) | | | | | | |
| | **Moderate or severe depression n (%)** | | **OR** | **(95% CI)** | **Adjusted OR 1** | **(95% CI)** | **Adjusted OR 2** | **(95% CI)** |
| | No | Yes | | | | | | |
| Disability status | | | | | | | | |
| None | 4280 (93) | 337 (7) | 1 | | 1 | | 1 | |
| At least some difficulty | 383 (76) | 122 (24) | 4.05 | 3.16 to 5.18 | 3.12 | 2.36 to 4.14 | 2.91 | 2.13 to 3.99 |
| All | 4663 (91) | 459 (9) | | | | | | |
| Physical or sexual IPV | | | | | | | | |
| No | 4186 (93) | 308 (7) | 1 | | 1 | | | |
| Yes | 477 (76) | 151 (24) | 4.30 | 3.40 to 5.44 | 3.97 | 3.10 to 5.09 | – | – |
| All | 4663 (91) | 459 (9) | | | | | | |
| Emotional IPV | | | | | | | | |
| No | 4204 (92) | 311 (7) | 1 | | 1 | | | |
| Yes | 459 (76) | 148 (24) | 4.36 | 3.40 to 5.58 | 4.01 | 3.04 to 5.29 | – | – |
| All | 4663 (91) | 459 (9) | | | | | | |
| | **Suicidal ideation n (%)** | | **OR** | **(95% CI)** | **Adjusted OR 1** | **(95% CI)** | **Adjusted OR 2** | **(95% CI)** |
| | No | Yes | | | | | | |
| Disability status | | | | | | | | |
| None | 4089 (89) | 528 (11) | 1 | | 1 | | 1 | |
| At least some difficulty | 384 (76) | 121 (24) | 2.44 | 1.98 to 3.01 | 2.13 | 1.66 to 2.75 | 1.94 | 1.50 to 2.50 |
| All | 4473 (87) | 649 (13) | | | | | | |
| Physical or sexual IPV | | | | | | | | |
| No | 4087 (91) | 407 (9) | 1 | | 1 | | | |
| Yes | 386 (61) | 242 (39) | 6.30 | 5.01 to 7.92 | 6.04 | 4.80 to 7.62 | – | – |
| All | 4473 (87) | 649 (13) | | | | | | |
| Emotional IPV | | | | | | | | |
| No | 4112 (91) | 403 (9) | 1 | | 1 | | | |
| Yes | 361 (59) | 246 (41) | 6.95 | 5.81 to 8.33 | 6.55 | 5.36 to 8.00 | – | – |
| All | 4473 (87) | 649 (13) | | | | | | |

Adjusted OR 1: adjusted for sociodemographic covariates: woman's age, marital status, schooling, employment, religion, socioeconomic quintile and husband's schooling, employment and alcohol or drug use.
Adjusted OR 2: adjusted for sociodemographic covariates and preceding year physical or sexual IPV and emotional IPV.
ORs and 95% CIs from univariable and multivariable logistic regression models.
CI, confidence interval; IPV, intimate partner violence; OR, odds ratio.

needs to be communicated to relevant stakeholders and methods to address it developed.

The intersection of poverty, inequality and discriminatory social norms is central to both violence and mental health disorders.[94 95] In visualising and categorising interconnected determinants, the ecological framework used in our analysis has utility. Determining the source of violence, whether at individual micro or broader macro levels, is less clear-cut. The effects of multidimensional poverty and gender in ecologies of violence have received attention in recent years,[10 96] but more research is needed, both quantitative and qualitative, to understand the place of disability and mental health in this matrix, particularly in the context of India. A potential data source is India's National Family Health Survey (NFHS), which has been expanded to include measures of disability.[97] The NFHS has already been used widely in IPV studies involving non-disabled women.[41 98] It is hoped that this change will bring greater visibility to women with disabilities in investigations of violence.

Alongside improved visibility, regressive social and cultural norms must be challenged. These include the conception of disability as a personal failing or a sign of ill fate, and of disabled people as deserving of pity,[99 100] as well as patriarchal and stigmatising attitudes towards women and mental disorder,[101 102] and tolerance of violence among both women and men.[103] This can be achieved, in part, through integrated interventions that incorporate antiviolence messages into health and educational activities, and through targeted interventions that seek to change violence-related norms directly.[56 104] These efforts work best when addressing both men and women across all social classes and levels of the social ecology.[67] A recent targeted pilot study in Jharkhand, India that mobilised community resources through participatory learning and action groups facilitated by Accredited Social Health Activists (ASHAs) reported reductions in experience and tolerance of violence and greater levels of help-seeking.[105] The study contributes to a growing body of evidence demonstrating that norms and practices that perpetuate inequities and violence can be challenged by community mobilisation and education.[106] Such interventions should work with disabled communities and activists, incorporating disability-positive messaging, to challenge discrimination at the community level and provide a more nuanced and inclusive approach to antiviolence efforts.

In healthcare settings, efforts should be made to identify and support disabled women unknown to healthcare and violence support services. Routine healthcare and IPV education and screening programmes need to be disability-inclusive, and service providers mindful of the vulnerability of disabled women. At screenings, disabled women should have access to safe and private spaces, away from accompanying family members who may be party to violence. Those delivering screening may need training to consider different kinds of functional difficulty and access needs such as adapted communication. Training

should also incorporate issues of trust and respect in order to challenge the stereotyping that can cause women with disabilities to be disbelieved when reporting episodes of violence.[100 107] The need to challenge stereotypes and promote respect extends to intersecting issues such as widowhood, separation and divorce. Women can be stigmatised in India after the loss of a husband or the ending of a marriage, with a burden of fault placed on them.[108] In our study and elsewhere,[11] women with disabilities were more likely to be separated or divorced, which is likely to further compound any stigma they already experience. Healthcare workers are well placed to challenge these kinds of stereotypes, but may themselves be party to sustaining them and require support to break reductive patterns of behaviour and thinking.

At a policy level, three major instruments in Indian law pertain directly to the evidence presented herein: the Protection of Women from Domestic Violence Act, 2005 (PWDVA),[109] the Rights of Persons with Disability Act, 2016[110] and the Mental Healthcare Act, 2017.[111] The greater scope for understanding and protection that these most recent instruments afford is a step in the right direction. The Rights of Persons with Disability Act brought India's disability policy in line with the United Nations Convention on the Rights of Persons with Disabilities and introduced a rights-based approach towards disability,[112] while the Mental Healthcare Act shifted the approach to mental health away from criminalisation towards provision of healthcare.[113] However, there were limitations in the development of both,[112 114] including the Mental Healthcare Act's failure to sufficiently address stigma.[115] As for the PWDVA, India's persistently high rates of domestic violence, which have grown during the COVID-19 pandemic,[116] are testament to the act's failure to effect any real change in society.[117 118] This further demonstrates that, while policy can be an important tool for addressing attitudinal and environmental barriers, change will remain elusive without commitment to implementation supported by adequate funding and monitoring-accountability mechanisms.

### Limitations

Limitations include the cross-sectional study design, which precludes determination of whether disability was an outcome of or risk factor for IPV. Use of self-report for exposure items is a further limitation, although the recall period was short, participants being asked about their experiences in the last year. Responses to questions about IPV and mental health may have been self-censored due to the sensitive nature of the questions and therefore underestimated, despite researchers being familiar figures in the community and their efforts to develop trust. Use of self-report for disability is also a limitation and may be reflected in the low rates. The WG-SS determines a person's limitations based on self-perception in relation to their environment. Given the disadvantaged setting in which the participants lived, this may have also impacted the disability rates.

## Conclusion

Violence against women in India is a major and ongoing public health problem. Women with disabilities are perhaps the most marginalised of all and experience violence to a greater degree than others. This violence derives from a culture that tolerates and perpetuates the marginalisation of both women in general and women with disabilities in particular and contributes to an unacceptable physical and mental health burden. The issue requires greater attention from national and community leaders to address its causes, especially poverty and gender inequality, and to meet India's commitments to its disabled population.

**Acknowledgements** We are grateful to the women and community guardians who agreed to contribute to the study. We thank the investigators, data collection supervisors Miheeka Vast and Manju Singh, Bhaskar Kakad for supervision, Gauri Savkur for training support, Unnati Machchhar and Shilpa Adelkar for subsequent supervision of the intervention programme, Archana Bagra and Vibhavari Bali for financial and human resources management, and Vanessa D'Souza and Shanti Pantvaidya for leadership at SNEHA.

**Contributors** AR, ND and DO conceived the study. ND, MW, SK, AG and DO developed the methodology for data collection. ND, SK and AG oversaw investigation. AR and DO curated the data and designed the analysis. AR did the analysis and wrote the first draft. ND managed the project and ND and DO acquired funding. All authors critically reviewed drafts of the manuscript and read and commented on the final version. AR is guarantor and corresponding author and accepts full responsibility for the work herein.

**Funding** This research was funded in part by the Wellcome Trust (206417). For the purpose of open access, the author has applied a CC BY public copyright licence to any Author Accepted Manuscript version arising from this submission.

**Disclaimer** The funder had no role in study design, data collection and analysis, decision to publish, or preparation of the manuscript.

**Competing interests** None declared.

**Patient and public involvement** Patients and/or the public were involved in the design, or conduct, or reporting, or dissemination plans of this research. Refer to the Methods section for further details.

**Patient consent for publication** Not applicable.

**Ethics approval** This study involves human participants and was approved by UCL Research Ethics Committee (3546/003 27/09/2017) PUKAR (Partners for Urban Knowledge, Action and Research) Institutional Ethics Committee (25/12/2017).

**Provenance and peer review** Not commissioned; externally peer reviewed.

**Data availability statement** Data are available in Open Science Framework a public, open access repository. https://osf.io/zhtpw/.

**ORCID iDs**
Andrew Riley http://orcid.org/0000-0001-5569-8694
Suman Kanougiya http://orcid.org/0000-0003-0007-3157
David Osrin http://orcid.org/0000-0001-9691-9684

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
