## [Reviewer comments · BMJ Open]

ARTICLE DETAILS

TITLE (PROVISIONAL)	Intimate partner violence against women with disability and associated mental health concerns: a cross-sectional survey in Mumbai, India
AUTHORS	Riley, Andrew; Daruwalla, Nayreen; Kanougiya, Suman; Gupta, Apoorwa; Wickenden, Mary; Osrin, David

VERSION 1 – REVIEW

REVIEWER	Ojeahere, Margaret Isioma Jos University Teaching Hospital, Psychiatry
REVIEW RETURNED	06-Oct-2021

GENERAL COMMENTS	The authors raised several interesting points around intimate partner violence, disability and mental illness. A pertinent study yet under-studied topic. I believe this is an important contribution to the literature base. Methods • Please provide more details about your recruitment process especially for the interviewers. The referenced study (Daruwalla N, Kanougiya S, Gupta A, Gram L, Osrin D. Prevalence of domestic violence against women in informal settlements in Mumbai, India: a cross-sectional survey. BMJ Open. 2020;10(12):e042444. Published 2020 Dec 16. doi:10.1136/bmjopen-2020-042444) states that “16 female interviewers with graduate education and 3 months of training visited households”.• How were they selected? Did they have any background in mental health? Did they have baseline training in handling people with disabilities? Were they graduates of some sort of special education or were they trained during the 3 months? How was this interview carried out for participants who had some degree of visual, hearing and speaking impairment? It would be helpful to explain a bit more.• Is it possible that the sensitive nature of the questions and the interviews conducted in the participants/respondents’ houses, some family members who stayed during the interview and the possibility of being overheard considering the living conditions (i.e walls have ears) may have affected the results beyond underestimation? Please briefly explain the implications of this sentence on the participants and how the authors ensured safeguarding them beyond the helplines viz a viz the ethical measures taken to ensure this.• Please provide examples of the survey questions (sociodemographic questions). It would be helpful to include a copy of the survey questions as an appendix with the article.
--

	 • Page 12, line 10 “Participants” is used interchangeably with “respondents”. Similarly seen on Page 18, line 3, 10, 24, Page 19, line 6 -“Participants was used here, although respondents had been used previously. I suggest that for clarity authors should use either respondents or participants. • The use of terms such as “approximately, about” e.t.c may be more appropriate, instead of the symbol “~”. However, the author may choose to disregard this. • Page 15, line 10-17. This sentence appears to fit better in either the introduction or discussion section. • Page 16, line 7 “At woman level”.... While this speaks to category variables which centers around the woman as a participant, it appears somewhat pejorative. I would suggest re-wording it perhaps “At respondent/participant level”. • Page 16 line 14-17 -I recommend including appropriate references to support this sentence. • Page 18. Line 49 I would suggest the use of “respondent or participant” rather than “patient”. • Page 19, line 31 “eligible woman resident” is unclear. This paragraph is unclear and would benefit from being re-worded. • Page 21, line 3 those who earned? it would be easier for the reader if the authors clarified this i.e “earned what”. Discussion  • Page 32. Line 17 “the ending of marriage”- this would benefit from being rephrased. • Page 32, line 22-28 I suggest this sentence be reworded for clarity. • I would have loved to see more discussion on the intersection between significant sociodemographic characteristics, mental illness, and other identified disabilities. • It would be helpful to expand upon the discussion as to why the authors think their prevalence rates differed from other studies. Limitations  • Page 36 line 56 up to page 37 to line 24- This part mentions prevalence and compares with other studies, it would support findings from the study. I recommend that this part be included in the discussion section. Recommendations  • I like the strong recommendations that originate from this study and are highlighted by the authors. However, the authors mentioned some recommendations while discussing the results of the study, early in the discussion. This disrupts the flow and makes it difficult to read. • I suggest that the authors should edit for clarity and the flow of the paper. I propose the authors revise and resubmit the manuscript.
--	--

REVIEWER	Ahmad, Noor Ani Institute for Public Health, National Institutes of Health, Ministry of Health Malaysia
REVIEW RETURNED	08-Oct-2021

GENERAL COMMENTS	Background: Page 6 line 45: should explain why the prevalence was considered as “underestimate” Page 8 para 2: to include the difference in terms were used: generalised violence, domestic violence, as part of limitations. No description on the relationship between IPV and mental health. Suggest to add in the introduction section. Methods
---

	Setting: based on the description can be considered as underprivileged population. Kindly clarify. Eligibility of the respondent or inclusion/exclusion criteria not mentioned anywhere in methods section. Limitation: To add the limitation of using WG Short set Questionnaire to determine people with difficulties as the tool assess the limitations based on their perception in relation to their environment; in particular the study setting with various environmental setbacks. Overall comments: please use Person-First Language throughout the manuscript. Kindly do not label people with the term “disabled ..”.
--	---

VERSION 1 – AUTHOR RESPONSE

Reviewer: 1

Methods

- Please provide more details about your recruitment process especially for the interviewers. How were they selected? Did they have any background in mental health? Did they have baseline training in handling people with disabilities? Were they graduates of some sort of special education or were they trained during the 3 months?

We have added to the methods section: “Sixteen women interviewers collected data. They were graduates who had not worked in the mental health field and were selected after application and interview through open recruitment. They received three months of training, which included three days of training facilitated by an expert on disability awareness and the Washington Group Short Set of questions.”

- How was this interview carried out for participants who had some degree of visual, hearing and speaking impairment?

We have added to the methods section: “For respondents with visual impairment, the interviews were oral, as they were for all respondents. For participants with learning difficulties, we developed an alternative interview in a simple visual format that was used 14 times during the survey.” The simple format version has been appended in an additional file.

- Is it possible that the sensitive nature of the questions and the interviews conducted in the participants/respondents’ houses, some family members who stayed during the interview and the possibility of being overheard considering the living conditions (i.e walls have ears) may have affected the results beyond underestimation? Please briefly explain the implications of this sentence on the participants and how the authors ensured safeguarding them beyond the helplines viz a viz the ethical measures taken to ensure this.

It is unlikely that the data were affected beyond underestimation through participant self-censorship, but we agree with the reviewer that the duty of care extends beyond providing helplines. The article by Daruwalla et al. cited above provides a detailed account of the safeguarding measures taken by the research team, which is quoted below. In the article, as we are already at the word limit, we have emphasised the efforts made to ensure privacy and that duty of care was paramount, and have reiterated that safeguarding details are published in the linked article.

“Interviewers were all women and provided both time and sufficient information for women to consider whether to participate. They were supported by three field supervisors with direct linkage to counselling services, available by phone at any time. To ensure privacy, interviews were arranged by advance appointment and avoided times when partners or children were likely to return from work or school. Women were interviewed at home or in a local community office if they preferred it. The interview began with general questions about demography, household residents, education, socioeconomic position, maternity and health. If a family member, neighbour or friend entered, the interviewer went back to asking questions about general health. If the person showed signs of staying, the interview was terminated and completed over up to three repeat visits. As a result of the gatekeeper consent process, community members were aware that interviewers would be visiting people in their area and this limited curiosity and intrusion.

“Of particular concern is the duty of care after disclosure. We believe that an interviewee who discloses experience of violence should be offered optimal support, and interviewers were members of a broader team who were able to provide a full suite of crisis and counselling services, including home visits, medical, surgical and psychiatric referral, and negotiation with families, the police and legal representatives. When survivors disclosed violence, we followed established intervention protocols which included safety assessment, counselling, liaison with healthcare, police and legal services, and developing follow-up plans for the survivor and her family. Participants were able to speak with counsellors immediately by phone. When a survivor was not ready to disclose violence, the interviewer provided her with information on available services and legal rights and gave her a small card that was easy to hide and listed essential contact numbers and addresses for 24-hour crisis support, medical emergencies and the police. She took consent for any action from the participant herself.”

- Please provide examples of the survey questions (sociodemographic questions). It would be helpful to include a copy of the survey questions as an appendix with the article.

- Example sociodemographic survey questions have been appended in Additional File 2.

- Page 12, line 10 “Participants” is used interchangeably with “respondents”. Similarly seen on Page 18, line 3, 10, 24, Page 19, line 6 -“Participants was used here, although respondents had been used previously. I suggest that for clarity authors should use either respondents or participants.

- Noted and amended to participant throughout.

- The use of terms such as “approximately, about” e.t.c may be more appropriate, instead of the symbol “~”. However, the author may choose to disregard this.

- We have removed the tilde symbol.

- Page 15, line 10-17. This sentence appears to fit better in either the introduction or discussion section.

- The sentence was added to illustrate the interplay of levels in Heise’s ecological framework. We agree that it disrupts the flow of this section and have removed it: the description of the framework is sufficient.

- Page 16, line 7 “At woman level”.... While this speaks to category variables which centers around the woman as a participant, it appears somewhat pejorative. I would suggest re-wording it perhaps “At respondent/participant level”.

This has been amended to 'at the level of individual women' to retain the link to the ecological framework which has been changed to refer to the 'individual' rather than 'woman'

• Page 16 line 14-17 -I recommend including appropriate references to support this sentence.

Noted and added.

• Page 18. Line 49 I would suggest the use of “respondent or participant” rather than “patient”.

'Patient' in this case is part of the phrase 'patient and public involvement', PPI, rather than referring to study participants. The section is required by the journal and the word is part of the journal mandate.

• Page 19, line 31 “eligible woman resident” is unclear. This paragraph is unclear and would benefit from being re-worded.

The paragraph has been reworded: 'Interviewers approached 5277 households between 5th December 2017 and 28th March 2019. In 967 (5%) households there was no eligible female resident; in 592 households (11%) the interviewers could not achieve privacy across repeat visits; and in 155 (3%) households a potential participant declined interview. In total, 5122 interviews were conducted.'

• Page 21, line 3 those who earned? it would be easier for the reader if the authors clarified this i.e “earned what”.

Amended to 'Having no remunerated work was common, while those who had an income ...'

Discussion

• Page 32. Line 17 “the ending of marriage”- this would benefit from being rephrased. Page 32, line 22-28 I suggest this sentence be reworded for clarity

This section has been revised.

• I would have loved to see more discussion on the intersection between significant sociodemographic characteristics, mental illness, and other identified disabilities.

As the reviewer implies, we do touch on this intersection, but only briefly. It is an important and complex issue that has not received the attention it needs. We do stress this need and point to a potential data source on which to build investigations. However, given the dearth of evidence and the word limit, we feel that this discussion should be taken up elsewhere.

• It would be helpful to expand upon the discussion as to why the authors think their prevalence rates differed from other studies.

A good point, on which we have expanded:

'The cause of this disparity is unclear. Strong efforts were made to help participants feel able to respond openly to questions, and the study used definitions of violence and survey instruments comparable with other studies: the lower rates may therefore be a fair reflection of levels of violence in this community. Indeed, the rate accords with a study of physical (12%), sexual (2%), and emotional IPV (8%) experienced six weeks postpartum also undertaken in Mumbai informal settlements [74].

'It is possible, nevertheless, that lower rates of violence may reflect under-reporting. Multiple factors play a role in the reporting of violence, such as a community's redressal structures, the extent of local welfare activities, and social norms. It may be that some communities are particularly close and not forthcoming enough to report violence. The context of each community is different and efforts must be made to understand local norms and behaviours in order to contextualise research outcomes.'

Limitations

- Page 36 line 56 up to page 37 to line 24- This part mentions prevalence and compares with other studies, it would support findings from the study. I recommend that this part be included in the discussion section.

We agree and have moved the section towards the start of the discussion.

Recommendations

- I like the strong recommendations that originate from this study and are highlighted by the authors. However, the authors mentioned some recommendations while discussing the results of the study, early in the discussion. This disrupts the flow and makes it difficult to read.

We agree and have moved the recommendations towards the end of the section.

- I suggest that the authors should edit for clarity and the flow of the paper.

We hope we have done this

- I propose the authors revise and resubmit the manuscript.

Accepted and done

Reviewer: 2

Background

- Page 6 line 45: should explain why the prevalence was considered as "underestimate"

Noted and added: they are believed to be underestimates 'due to proxy reporting and insufficient consideration of disability'

- Page 8 para 2: to include the difference in terms were used: generalised violence, domestic violence, as part of limitations.

The difference in category of violence in itself is not a limitation. However, we do concede that there is imprecision and a lack of detail in their definitions, which we have now acknowledged.

- No description on the relationship between IPV and mental health. Suggest to add in the introduction section.

The link between violence and mental health, with citations, is made in the penultimate paragraph of the background. Due to the constraint of the word limit, unfortunately we are unable to expand on the topic in this section. However, the issue is treated more fully in the discussion.

Methods

• Setting: based on the description can be considered as underprivileged population. Kindly clarify.

Noted and the adjective 'disadvantaged' added

• Eligibility of the respondent or inclusion/exclusion criteria not mentioned anywhere in methods section.

Noted and amended. The text now reads:

Inclusion criteria: Criteria for inclusion were women aged 18-49 years, one per household, resident in a study cluster, who consented to interview.

Limitation

• To add the limitation of using WG Short set Questionnaire to determine people with difficulties as the tool assess the limitations based on their perception in relation to their environment; in particular the study setting with various environmental setbacks.

Agreed and amended. The direction of effect the disadvantaged setting might have on reporting via the WG-SS is not clear cut. Living in an environment where the majority of people are disadvantaged may cause a person with disability to feel that they have a more equal status with others around them, compared to someone living in a more advantaged setting; or not. Unfortunately, there isn't room to explore this interesting effect in the limitations, so we have included the recommendation only.

Overall comments

• please use Person-First Language throughout the manuscript. Kindly do not label people with the term "disabled ...".

In this paper our approach to disability is in line with the social model which identifies disability as a social creation, 'a relationship between people with impairment and a disabling society' (Shakespeare, T., 2006. The Social Model of Disability. The Disability Studies Reader, 2, pp.197-204), thus distinguished from medical and individual models of disability. We therefore use the term 'disabled women', which denotes this position, interchangeably with 'women with disabilities', and do not consider the label to be pejorative. A footnote indicating this has been added.

VERSION 2 – REVIEW

REVIEWER	Ojeahere, Margaret Isioma Jos University Teaching Hospital, Psychiatry
REVIEW RETURNED	28-Dec-2021
GENERAL COMMENTS	*4th paragraph, line 2 in the discussion section -"The intersection of poverty, inequality, and discriminatory social norms is central to

	both violence and mental health disorder". Do the authors means "mental health disorders"? The authors have addressed the issues previously raised and have improved the quality of this paper. I believe this study would fill a gap and add to the body of knowledge, especially as it pertains to a subset of marginalized populations (people living with disabilities).
--	--

VERSION 2 – AUTHOR RESPONSE

Thank you for your help with our submission. Thanks also to our reviewer, Dr Ojeahere, for her statement of support. We have accepted and addressed all the points raised in your decision letter.

- We have amended the Strobe checklist as requested
- The 3rd bullet point has been removed from the Strengths and Limitations
- The supplementary file has been removed
- The grammatical point raised by Dr Ojeahere has been amended